# Significance of Histidine Hydrogen–Deuterium Exchange Mass Spectrometry in Protein Structural Biology

**DOI:** 10.3390/biology13010037

**Published:** 2024-01-09

**Authors:** Masaru Miyagi, Takashi Nakazawa

**Affiliations:** 1Department of Pharmacology, Case Western Reserve University, 10900 Euclid Avenue, Cleveland, OH 44106-4988, USA; 2Department of Chemistry, Nara Women’s University, Nara 630-8506, Japan

**Keywords:** histidine, histidine protonation, acid dissociation constant, mass spectrometry, protein structural biology

## Abstract

**Simple Summary:**

Various techniques are at our disposal for probing the structure and dynamics of proteins. Mass spectrometry stands out prominently in this realm due to its exceptional detection specificity and sensitivity. This enables the precise identification of individual molecular species in complex mixtures with remarkable sensitivity. The unique strengths of mass spectrometry have spurred the development of numerous approaches for characterizing protein molecules. In this review, our focus is on a specific mass-spectrometry-based method known as histidine hydrogen–deuterium exchange mass spectrometry (His-HDX-MS). We delve into its principle, experimental workflow, applications and the details of data analysis and interpretation.

**Abstract:**

Histidine residues play crucial roles in shaping the function and structure of proteins due to their unique ability to act as both acids and bases. In other words, they can serve as proton donors and acceptors at physiological pH. This exceptional property is attributed to the side-chain imidazole ring of histidine residues. Consequently, determining the acid-base dissociation constant (*K*a) of histidine imidazole rings in proteins often yields valuable insights into protein functions. Significant efforts have been dedicated to measuring the p*K*a values of histidine residues in various proteins, with nuclear magnetic resonance (NMR) spectroscopy being the most commonly used technique. However, NMR-based methods encounter challenges in assigning signals to individual imidazole rings and require a substantial amount of proteins. To address these issues associated with NMR-based approaches, a mass-spectrometry-based method known as histidine hydrogen–deuterium exchange mass spectrometry (His-HDX-MS) has been developed. This technique not only determines the p*K*a values of histidine imidazole groups but also quantifies their solvent accessibility. His-HDX-MS has proven effective across diverse proteins, showcasing its utility. This review aims to clarify the fundamental principles of His-HDX-MS, detail the experimental workflow, explain data analysis procedures and provide guidance for interpreting the obtained results.

## 1. Introduction

Histidine (His) residues exhibit exceptionally diverse features in protein functions, including catalysis, hydrogen bonding and metal binding. The distinctive chemical properties of His residues stem from their side-chain imidazole group, which bears two nitrogen atoms at N-1 (δ2) and N-3 (δ1) positions (Figure 1) [1]. These nitrogen atoms play crucial roles in the functions of His residues, serving either as hydrogen acceptors or donors in the two tautomeric forms of the imidazole group (Im), depending on the position of the hydrogen atom attached to either the N-1 or N-3 position. Importantly, either form of the imidazole group can act as a base and acquire a proton to become the cationic imidazolium group ($Im\cdot H^{+}$), which is stabilized by the resonance structure between its two tautomeric forms (Figure 1).

The dissociation constant of this acid-base equilibrium, $K_{a}$, is expressed in the following equation:(1)$$K_{a}=\frac{\left[Im\right]\left[H^{+}\right]}{\left[Im\cdot H^{+}\right]}$$
where $\left[Im\right]$, $\left[H^{+}\right]$ and $\left[Im\cdot H^{+}\right]$ denote the concentrations of imidazole, proton and imidazolium, respectively. For simplification, we often use the $pK_{a}$ value, which directly correlates with the pH of an aqueous solution. This connection is established by taking the logarithm of both sides of Equation (1), as shown below:(2)$$pK_{a}={-\log}_{10}\frac{\left[Im\right]}{\left[Im\cdot H^{+}\right]}+pH$$

The p*K*a value of the imidazole group in proteins can vary widely (from approximately 4 to 9) depending on its electrostatic environment [2,3,4]. This distinctive property enables His residue to engage in diverse catalytic processes, functioning either as an acid or a base. Even within a single catalytic process, the imidazole groups are capable of functioning as both proton donors (acid) and acceptors (base) in different catalytic steps, as seen in the catalysis of serine proteases [5,6,7]. Furthermore, this unique property facilitates the binding of His residues with metals over a considerably wide pH range. Consequently, the precise measurement of $pK_{a}$ values in His residues is often crucial to understanding the mechanism of enzyme reactions.

Until recently, nuclear magnetic resonance (NMR) spectroscopy has been by far the most widely used technique for determining the $pK_{a}$ values of His residues in proteins [5,8,9]. However, NMR-based approaches have significant drawbacks, including difficulties in assigning signals to individual His residues in proteins and the requirement for a large quantity of proteins. To address the challenge of signal assignment and detection sensitivity, a mass-spectrometry (MS)-based method called histidine hydrogen–deuterium exchange mass spectrometry (His-HDX-MS) was developed in 2008 [10]. Since then, His-HDX-MS has been demonstrated to overcome the limitations of NMR-based approaches and has proven to be useful not only in determining the $pK_{a}$ values of His residues but also in probing the conformational changes in proteins.

In this review, we begin by elucidating the foundational principles of His-HDX-MS. Subsequently, we detail the experimental workflow, explore the applications of this method, outline data analysis procedures and provide guidance on interpreting the results.

## 2. $pK_{a}$ Value and Kinetic Parameters of the Acid-Base Equilibrium of His Residues in Proteins

### 2.1. Measurement of $pK_{a}$ with NMR Spectroscopy

Until recently, NMR spectroscopy has been by far the most widely used technique for determining the $pK_{a}$ values of His residues in proteins [5,8,9,11,12]. The most advantageous feature of NMR is that the ^1^H and ^13^C signals of the imidazole C-2 position (Figure 1) are well resolved from the majority of other resonances, even in one-dimensional spectra. The chemical shift in C-2 signal observed at δ_obs_ (ppm) reflects the weighted average of limiting chemical shits $\delta_{Im}$ and $\delta_{Im\cdot H+}$ of the imidazole group in non-protonated (Im) and protonated ($Im\cdot H^{+}$) forms, respectively (Figure 2). The differences in chemical shifts $\delta_{Im\cdot H+}-\delta_{obs}$ and $\delta_{obs}-\delta_{Im}$ are proportional to concentrations [Im] and $\left[Im\cdot H^{+}\right]$, respectively. Therefore, $pK_{a}$ can be expressed in an equation replacing the $\left[Im\cdot H^{+}\right]$/[Im] term of the Henderson–Hasselbalch equation (Equation (2)) with $\left(\delta_{ImH+}-\delta_{obs}\right)$/$\left(\delta_{obs}-\delta_{Im}\right)$, as follows:(3)$$pK_{a}=-\log_{10}\frac{\left(\delta_{Im\cdot H+}-\delta_{obs}\right)}{\left(\delta_{obs}-\delta_{Im}\right)}+pH$$

Although this method is simple and easy to perform, difficulties persist in unambiguously assigning ^1^H, ^13^C and ^15^N signals to individual C-2 resonances, even in a small protein containing only a few His residues. In principle, it is still possible to assign every C-2 signal to the relevant His residue using sophisticated multi-dimensional NMR techniques. For example, the $pK_{a}$ values of seven His residues in human carbonic anhydrase II (29 kDa) have been determined via pH titration of selectively enriched ^15^N signals of imidazole N-1 and N-3 atoms, which were assigned by a combination of multi-dimensional heteronuclear correlation spectroscopy [13]. However, the necessity to prepare ^13^C- and ^15^N-labeled recombinant proteins at a considerably high concentration (1–3 mM) represents a significant limitation of the method’s applicability to larger proteins [8,13]. Furthermore, preparing isotope-labeled large proteins in large quantities is expensive and time consuming.

### 2.2. Measurement of $pK_{a}$ and Kinetic Parameters Using Tritium Labeling

Prior to the invention of electrospray ionization (ESI) mass spectrometry, tremendous efforts were made by Matsuo and colleagues to address the drawbacks of NMR-based approaches using tritium (^3^H or T) labeling [14,15]. In this method, proteins were incubated in buffers at various pH values containing tritiated water (^3^H_2_O) for ~2 days. Subsequently, the proteins were digested using a combination of proteases, and the resulting peptides were separated through two-dimensional paper chromatography electrophoresis. The radioactive spots were then excised, and their radioactivities were measured. The identity of the radioactive peptides was determined through amino acid analysis. Although this tritium labeling method based on hydrogen–tritium exchange (HTX) reaction successfully alleviated the problems of NMR experiments by achieving much higher sensitivity and the accurate assignment of tritium-incorporated His residues, its technical complexity was daunting to the majority of researchers, except for a few peptide chemists. In fact, a seminal textbook on protein science referred to this method as follows: “These experiments are tedious to perform, but the assignments of $pK_{a}$ are unambiguous” [16].

### 2.3. Measurement of $pK_{a}$ and Kinetic Parameters Using MS

To address the challenges associated with NMR signal assignment, Miyagi and Nakazawa proposed employing ESI mass spectrometry [10]. Their successful demonstration revealed that this mass-spectrometry-based approach effectively resolves signal assignment issues encountered in NMR spectroscopy and significantly reduces the quantity of protein required. In the following sections, we delve into the mechanism of HDX reaction at the imidazole ring of His residues (Section 2.3.1), the calculation methods for obtaining the HDX rate constant from the acquired mass spectrum (Section 2.3.2) and the relationship between $pK_{a}$ values and HDX rates (Section 2.3.3).

#### 2.3.1. The Mechanism of HDX Reaction at the C-2 Position of the Imidazole Group

Hydrogens, which are covalently attached to nitrogen, oxygen or sulfur atoms, exchange rapidly with deuterium atoms in D_2_O solvent. Additionally, the C-2 hydrogen of the imidazole ring of His residues exchanges with deuterium but at a much slower rate, with a half-life in the order of days at ambient temperature [17,18]. The C-2 hydrogen is the only hydrogen in proteins, which is covalently bonded to a carbon atom but still exchanges with the hydrogen of aqueous solvents.

The HDX reaction at the C-2 position of the His imidazole ring has been established as a base-catalyzed process involving the abstraction of a proton from the C-2 position of the imidazolium ion [$Im\cdot D^{+}$] to form an ylide in the rate-determining step [19,20,21,22] (Figure 2). Therefore, the rate of reaction can be expressed in the following equation using the second-order rate constant, $k_{2}$, and concentrations of the imidazolium ion [$Im\cdot D^{+}$] and deuteroxide ion $\left[{OD}^{-}\right]$.
(4)$$rate=k_{2}\left[{Im\cdot D}^{+}\right]\left[{OD}^{-}\right]$$

Given that the overall reaction adheres to pseudo-first-order kinetics [20], and the rate is solely dependent on the concentration of the imidazole group, $\left[{Im}_{total}\right]$, the rate equation can also be expressed as
(5)$$rate=k_{\varphi }\left[{Im}_{total}\right]$$
where $k_{\varphi }$ is the pseudo-first-order rate constant. Rearrangement of Equations (4) and (5) yields the following expression:(6)$$k_{\varphi }=\frac{k_{2}K_{W}}{\left(K_{a}+\left[D^{+}\right]\right)}$$
Here, $K_{a}$ represents the dissociation constant of imidazolium cation ($K_{a}=\left[Im\right]\left[D^{+}\right]/\left[Im\cdot D^{+}\right]$), and $K_{W}$ represents the ion product of water ($K_{W}=\left[D^{+}\right]\left[{OD}^{-}\right]$), with a value of 10^−14.87^ [23]. As the pH increases, the value of $k_{\varphi }$ approaches its maximum value, $k_{\varphi }^{max}$. Under conditions where $K_{a}$ >> $\left[D^{+}\right]$, $k_{\varphi }^{max}$ is expressed by the equation
(7)$$k_{\varphi }^{max}=\frac{k_{2}K_{W}}{K_{a}}$$

Using Equation (7) with experimentally obtained $pK_{a}$ and $k_{\varphi }^{max}$ values, one can estimate the second-order rate constant $k_{2}$. Plugging the estimated value of $k_{2}$ with varying concentrations of $\left[D^{+}\right]$ into Equation (6) allows one to simulate the theoretical HDX titration curve of a given imidazole ring. Figure 3 shows the ribonuclease A (RNase A) His12 titration curve obtained previously (blue line) [10] and its theoretical curve calculated using Equation (6) (red line), demonstrating that the experimental data and theoretical data based on the reaction mechanism agree well. 

#### 2.3.2. Calculation of the Pseudo-First-Order Rate Constant $k_{\varphi }$ from a Mass Spectrum

The pseudo-first-order rate constant $k_{\varphi }$ can be obtained directly from the acquired mass spectrum. Figure 4 shows the hypothetical mass spectra of a peptide containing one histidine residue before (red) and after (blue) HDX reaction. The intensities of M and M + 1 peaks before HDX reaction, $I_{M}\left(0\right)$ and $I_{M+1}\left(0\right)$, respectively, could change to $I_{M}\left(t\right)$ and $I_{M+1}\left(t\right)$ after the reaction incorporating deuterium. Since the HDX reaction follows first-order kinetics, the changes in $I_{M}$ and $I_{M+1}$ peaks over time (after time *t*) can be expressed by Equations (8) and (9) [10].
(8)$$I_{M}\left(t\right)=I_{M}\left(0\right){\cdot e}^{-k_{\varphi }t}$$
(9)$$I_{M+1}\left(t\right)=I_{M+1}\left(0\right){\cdot e}^{-k_{\varphi }t}+I_{M}\left(0\right)\left[1-e^{-k_{\varphi }t}\right]$$

Dividing Equation (9) by Equation (8) yields
(10)$$\frac{I_{M+1}\left(t\right)}{I_{M}\left(t\right)}=\frac{I_{M+1}\left(0\right)}{I_{M}\left(0\right)}+e^{k_{\varphi }t}-1$$
Since *I*_M+1_(0)/*I*_M_(0) and *I*_M+1_(*t*)/*I*_M_(*t*) represent the ratios between $I_{M}$ and $I_{M+1}$ peaks before (time = 0) and after (time = *t*) HDX reaction, respectively, defining these ratios as *R*(0) and *R*(*t*), Equation (10) can be expressed as follows:(11)$$R(t)=R(0)+e^{k_{\varphi }t}-1$$
Taking the logarithm on both sides of Equation (11) and rearranging it yields the following equation:(12)$$ln\left[1+R\left(t\right)-R\left(0\right)\right]=k_{\varphi }t$$
This equation expresses the linear relationship between $ln\left[1+R\left(t\right)-R\left(0\right)\right]$ and time *t* with the slope as $k_{\varphi }$. Therefore, $k_{\varphi }$ can be calculated using the following equation.
(13)$$k_{\varphi }=ln\left[1+R(t)-R(0)\right]/t$$
Once the value of $k_{\varphi }$ is obtained, it is possible to estimate $pK_{a}$ and $k_{\varphi }^{max}$ values graphically using a non-linear least-square fit program (see Figure 3). 

In the HDX experiments using a mixture of H_2_O and D_2_O as solvent, Equations (8) and (9) should be modified to include the fraction of D_2_O content in the solvent, $p=\left[D_{2}O\right]/\left(\left[D_{2}O\right]+\left[H_{2}O\right]\right)$, as a factor limiting the maximum increase in mass by p Da. That is, transforming Equations (8) and (9) to
(14)$$I_{M}\left(t\right)=I_{M}\left(0\right)-\left[I_{M}\left(0\right)\left(1-e^{-k_{\varphi }t}\right)p\right]$$
and
(15)$$I_{M+1}\left(t\right)=\left\{I_{M+1}(0)-\left[I_{M+1}(0)\left(1-e^{-k_{\varphi }t}\right)p\right]\right\}+\left[I_{M}\left(0\right)\left(1-e^{-k_{\varphi }t}\right)p\right]$$
From Equations (14) and (15), the following equation for obtaining $k_{\varphi }$ is derived [24].
(16)$$k_{\varphi }=-ln\left\{1-\left[\frac{\left\{R\left(t\right)-R\left(0\right)\right\}}{1+R\left(t\right)-R\left(0\right)}\right]\frac{1}{p}\right\}/t$$
Equation (13) is considered a special case of Equation (15), in which *p* =1 (100% D_2_O). Equation (16) is particularly useful when a mixed solvent of D_2_O and H_2_O is used in the HDX reaction. 

It is also possible to obtain $k_{\varphi }$ by employing the shift in weighted average mass $M_{wt,av}\left(t\right)$ in the range $M_{wt,av}\left(0\right)\leq M_{wt,av}\left(t\right)\leq M_{wt,av}\left(0\right)+1$ or $0\leq {∆M}_{wt,av}\left(t\right)\leq 1$ [${∆M}_{wt,av}\left(t\right)=M_{wt,av}\left(t\right)-M_{wt,av}\left(0\right)$] [25,26]. The values $M_{wt,av}\left(0\right)$ at *t* = 0 and $M_{wt,av}\left(t\right)$ at time *t* can be calculated from the isotopomer distribution of a given peptide peak. Since the HDX reaction follows pseudo-first-order kinetics, the shift in weighted average mass over time (*t*) can be expressed as follows:(17)$$ln\left[1-∆M_{wt,av}\left(t\right)\right]={-k}_{\varphi }t$$
Therefore, $k_{\varphi }$ can be calculated using the following equation.
(18)$$k_{\varphi }=-ln\left[1-∆M_{wt,av}\left(t\right)\right]/t$$
Equation (17) can incorporate the factor *p* (the fraction of D_2_O content in the reaction) by replacing the maximum $M_{wt,av}\left(t\right)=1$ with p, as follows:(19)$$ln\left[p-∆M_{wt,av}\left(t\right)\right]={-k}_{\varphi }t$$
Thus, $k_{\varphi }$ can be calculated using the following equation.
(20)$$k_{\varphi }=-ln\left[p-∆M_{wt,av}\left(t\right)\right]/t$$

When the isotopic peak envelope of a molecular ion is obtained with a good signal-to-noise (S/N) ratio and without any interference peaks, Equations (13) and (18) can be reliably used without any preference. Regardless of the choice, it is essential to exercise extra caution for possible errors caused by low S/N spectra or the presence of interference peaks. Elferich and co-workers demonstrated that accurate calculation of deuterium incorporation can be achieved even with low S/N data by fitting the observed isotopic distribution with a linear combination of theoretical isotopic distributions of non-exchanged and fully exchanged peptides [27]. Another consideration to keep in mind when analyzing HDX data is that in His-HDX experiments, the protein samples are typically incubated for an extended period (more than 2 days). Therefore, special caution should be exercised when analyzing the mass spectrometry data, as deamidation of Asn or Gln residues could occur, resulting in an additional increase in mass of 1 Da. Since liquid chromatography is likely to separate the original peptide and its deamidated forms, LC-MS would be less susceptible to this problem than MALDI-MS.

#### 2.3.3. Relationship between $pK_{a}$ and $k_{\varphi }^{max}$

The maximum hydrogen–deuterium exchange (HDX) rates, represented as intrinsic $k_{\varphi }^{max}$ (kφmaxi), for C-2 hydrogens fully exposed to the solvent vary based on their $pK_{a}$ values, as illustrated in Figure 5. Consequently, for a precise quantitative interpretation of the experimentally obtained $k_{\varphi }^{max}$ value for a particular His residue, it is essential to be aware of the kφmaxi value of an imidazole group sharing the same $pK_{a}$ value as the given imidazole group.

The maximum exchange rates of C-2 hydrogens, which are fully exposed to the solvent—defined as intrinsic maximum HDX rates, kφmaxi—differ depending on their $pK_{a}$ values, as shown in Figure 5 [28]. Therefore, to quantitatively interpret the experimentally obtained $k_{\varphi }^{max}$ value for a specific His residue, it is necessary to know the kφmaxi value of an imidazole group with the same $pK_{a}$ value as the given imidazole group. 

As shown in Figure 2, the hydrogen exchange reaction of imidazole C-2 position is based on a base-catalyzed reaction to which a linear free-energy relationship known as the Brønsted equation is applicable, as follows:(21)logk=α·pKa+β

Matsuo and colleagues determined factor *α* to be –0.7 using the hydrogen–tritium exchange (HTX) method, with several model compounds containing an imidazole group fully exposed to water to correlate $k_{\varphi }^{max}$ with accessibility of the imidazole group to the solvent [29,30,31]. This linear relation still proved useful in the HDX method applied to His residues in RNase A [20]. 

Miyagi and colleagues also determined factor α and β using four imidazole derivatives, shown in Figure 5, resulting in the following equation [28]:(22)logkφmaxi=0.316pKa−7+0.057T−3.747

This equation involves temperature *T* in the Celsius scale for the sake of convenience in establishing the relationship between kφmaxi and $pK_{a}$ values at any temperature. The inversion of the logarithm function yields the kφmaxi value. By utilizing the protection factor (PF), defined as follows, one can quantitatively estimate the solvent accessibility of His residues in proteins [32].
(23)PF=kφmaxikφmax

In this context, a smaller PF value corresponds to higher solvent accessibility, with a value of 1 indicating complete exposure to the bulk solvent.

## 3. Experimental Workflow of His-HDX-MS Experiment

The general workflow of the His-HDX-MS experiment is depicted in Figure 6. First, the protein(s) is/are incubated in buffers made with D_2_O at various pD values for 2–3 days to allow His residues to incorporate a deuterium atom. After incubation, the reaction is quenched by adding formic acid to adjust the solution’s pH to below 4. Then, the protein is digested in H_2_O, during which all the deuterium atoms incorporated into amide (Asn/Gln/backbone), OH (Ser/Thr), NH_2_ (Lys/N-terminal), COOH (Asp/Glu/C-terminal), indole NH (Trp), ε-, η_1_- and η_2_-NH (Arg), SH (Cys) and imidazole NH (His) exchange back to protium almost instantaneously (<1 s at physiological pH at 37 °C, if not protected from access to H_2_O) [33], leaving the deuterium atom at the imidazole C-2 position (C-2 D) as the only deuterium remaining in the protein. To minimize the extent of C-2 D to C-2 H back-exchange reaction, the digestion is often completed within 1–2 h using immobilized proteases at pH 7–8. However, even when digesting proteins for 16 h at 37 °C, only approximately 16% of C-2 D is exchanged back to C-2 H based on the rate constant for the back-exchange reaction (k=0.011 h^−1^) [34]. After digestion, the digests are analyzed by LC-MS/MS. The back-exchange reaction during a typical LC-MS analysis under acidic pH condition is negligible, since essentially no C-2 D to C-2 H reaction occurs below pH 4 [10]. The mass spectrometry data obtained are then analyzed to obtain the pseudo-first-order rate constant ($k_{\varphi }$), as discussed above. The rate of HDX reaction ($k_{\varphi }$) as a function of pD yields a sigmoidal curve. The rate profile provides two useful parameters, which indicate the local environment of the given imidazole group in a protein. The first is the $pK_{a}$ of the imidazole N–H group, which coincides with the inflection point of the sigmoidal curve. The second is the maximum pseudo-first-order rate constant, $k_{\varphi }^{max}$, which corresponds to the upper plateau of the sigmoidal curve and indicates the solvent accessibility of the imidazole group expressed as a PF value, as discussed above. Based on the $k_{\varphi }^{max}$ value, the half-life of the HDX rate can also be calculated using the following equation: (24)$$t_{1/2}=\ln2/k_{\varphi }^{max}$$

The pH titration experiment detailed above represents a fully executed His-HDX-MS study, yielding both $pK_{a}$ and $k_{\varphi }^{max}$ values. Nonetheless, a simpler approach—such as measuring the HDX rates of His residues in unliganded and ligand-bound proteins at a fixed pH—can still offer significant insights into understanding the structural changes in proteins. However, it is crucial to highlight that the derived rate constant kφ is distinct from $k_{\varphi }^{max}$. While $k_{\varphi }^{max}$ allows for a quantitative description of solvent accessibility, the rate constant kφ simply represents a specific rate at the measured pH.

It should be noted, however, that obtaining residue-specific $k_{\varphi }$ or $k_{\varphi }^{max}$ values is not straightforward when multiple His residues exist in a peptide. This complexity arises because the precursor ion of the peptide reflects the combined HDX reactions from multiple His residues. To address this, careful selection of the protease(s) is necessary during protein digestion to ensure the production of peptides containing only a single His residue. An alternative way of tackling this issue is to use tandem mass spectrometry techniques. Wimalasena and colleagues used the collision-induced dissociation (CID) technique for a peptide with three His residues. They successfully determined the HDX rates of these three His residues using fragment ions containing only one of the three His residues [35]. Elferich and colleagues employed the electron-transfer dissociation (ETD) technique to fragment the furin propeptide (approximately 9200 Da) and successfully determined the $pK_{a}$ and $k_{\varphi }^{max}$ values of five His residues without using any proteases [27]. The employment of such peptide fragmentation techniques can significantly augment the effectiveness of His-HDX-MS, particularly for proteins with multiple His residues in a short sequence range. These examples illustrate the absence of C2-deuterium migration to other peptide regions during CID or ETD. This stands in contrast to amide-HDX-MS, where notable scrambling of amide-deuterons occurs during CID [36,37], albeit to a much smaller extent during ETD [38,39,40] or ultraviolet photodissociation (UVPD) [41,42].

## 4. Strengths and Limitations of His-HDX-MS

A variety of techniques are available for characterizing protein structures and dynamics. X-ray crystallography, NMR spectroscopy and cryo-electron microscopy are commonly chosen methods for determining protein structures at atomic resolution. Additionally, techniques such as circular dichroism spectroscopy, fluorescence spectroscopy and mass spectrometry are primarily employed to characterize the dynamic properties of proteins, focusing on backbone and side-chain mobilities. The strengths of mass spectrometry include (1) straightforward assignment of molecular ions to protein sequences, (2) high detection sensitivity and (3) no limitation on the sizes of proteins. Thus, compared to NMR-based methods, His-HDX-MS, which is based on mass spectrometry, provides a more straightforward assignment of $pK_{a}$ values to individual His residues, requiring a significantly smaller quantity of protein.

The most commonly used mass spectrometry technique when studying protein structure and dynamics is amide-HDX-MS [43,44,45], which monitors the HDX rates of the backbone amide. The major technical challenge of amide-HDX-MS is the fast back exchange from deuterium to proton, which can occur during protein digestion and mass spectrometry analysis. This reaction is fast; the half-life for the back-exchange reaction is about 1 h at pH 2.5 and at 0 °C [46,47], necessitating completion of the digestion of proteins at acidic pH and analysis of the resulting peptides with LC-MS/MS in approximately 1 h. In contrast, the rate of His-HDX is much slower (*t*_1/2_ = 1.9 days at 37 °C) [34], allowing for relatively long protein digestion at physiological pH at 37 °C using proteases, which work at physiological pH. Additionally, in amide-HDX-MS, the LC-MS separation of peptides is conducted at low temperatures (close to 0 °C) to avoid back exchange. However, since essentially no back-exchange reaction occurs below pH 4 [10], in His-HDX-MS, a long chromatographic separation of peptides at a higher temperature can be employed to enhance the efficiency of peptide separation [48,49]. 

The major limitations of His-HDX compared to amide-HDX-MS are twofold: (1) the technique relies on the presence of His residues and (2) requires a long incubation time (>48 h at 37 °C). The occurrence of histidine residues in human proteins is 2.6%, which equates to 2–3 His residues in every 100 amino acid residues [50]. Therefore, His-HDX-MS is limited to probing the local electrostatic environment and solvent accessibility of His residues. Another limitation is the requirement for a long incubation time. Thus, this technique is not suitable for proteins, which are not stable.

## 5. Applications of His-HDX-MS and the Interpretation of His-HDX-MS Data 

### 5.1. Applications of His-HDX-MS

The primary application of His-HDX-MS is in determining the $pK_{a}$ values and solvent accessibilities of histidine (His) residues within proteins. In a study by Miyagi and colleagues, changes in the $pK_{a}$ and $k_{\varphi }^{max}$ values of five histidine residues in dihydrofolate reductase (DHFR) upon ligand binding were investigated [24]. The findings underscored the high sensitivity of $pK_{a}$ and $k_{\varphi }^{max}$ values to changes in the electrostatic environment and solvent accessibilities of these histidine residues, respectively. A study by Hayashi and colleagues on RNase A in the presence and absence of its ligand—cytidine 3′-monophosphate (3′-CMP)—also provided in-depth insights into the micro-environment of three His residues, including possible hydrogen bonding networks [26]. Consequently, the determination of $pK_{a}$ and $k_{\varphi }^{max}$ values could provide valuable insights into the micro-environment of His residues, facilitating a better understanding of their roles in protein functions.

Elferich and colleagues determined the p*K*a values of the conserved His residue in the propeptides of furin and proprotein convertase 1/3 to be 6.0 and 5.6, respectively [27]. The result was consistent with the different activation pHs of these proteins (furin: pH 6.5 and proprotein convertase 1/3: pH 5.5), indicating that protonation of the conserved His residue triggers the activation processes of these proteins. Miyahara and colleagues investigated the susceptibility of monoclonal antibodies—adalimumab and rituximab—to photo-oxidation and found that photo-oxidation occurs preferentially in His residues with low $pK_{a}$ values [51], indicating that the unprotonated imidazole ring is more susceptible to oxidation. Therefore, storing antibodies under acidic conditions could prevent oxidation. The two studies underscore the fact that assessing the protonation states of His residues could provide insights into the pH-dependent conformational changes in proteins or susceptibility to photo-oxidation.

His-HDX-MS has been employed to study structural changes in the G-protein-coupled receptor rhodopsin in its apo and ligand-bound forms in the native membrane prepared from bovine retina [52]. This technique has also been used to study the activation mechanism of protease-activated receptor 4, another G-protein-coupled receptor [53]. These studies demonstrated that His-HDX-MS can be applied to membrane proteins without much difficulty, unlike amide-HDX-MS, which requires the digestion and analysis of protein samples within a short period of time (<2 h) due to the fast back-exchange reaction of amide deuteron to proton.

Surewicz’s group employed His-HDX-MS in conjunction with amide-HDX-MS to study the structures of different forms of prion proteins [54,55,56,57]. These studies showcased the complementary nature of His-HDX-MS and amide-HDX-MS. The former delves into the micro-environment of His residues, while the latter unveils the conformational flexibility of backbone amides throughout the structure. Therefore, it is recommended to leverage both techniques when technical expertise is available.

His-HDX-MS has also served as the basis for measuring the thermodynamic stability of proteins. Fitzgerald’s group pioneered the development of a method for assessing the thermodynamic stability of protein–ligand and protein–protein interactions [25,58]. In this approach, proteins are incubated in a D_2_O buffer with varying concentrations of a protein denaturant (e.g., guanidine hydrochloride), and proteins are then digested and analyzed by LC-MS/MS. The results provide protein-denaturant concentration-dependent HDX rates of His residues from which the free-energy change (ΔG°) in protein denaturation can be estimated. This approach has also been employed in monitoring the thermodynamic stability changes in anthrax protective antigen upon binding to its receptor—capillary morphogenesis protein-2 (CMG2)—revealing that the protective antigen is stabilized by CMG-2 [59]. Additionally, the approach has been utilized to identify protein–drug interactions [34]. Despite the availability of various techniques for measuring protein stability or protein folding/unfolding, the strength of His-HDX-MS lies in its applicability to complex protein mixtures.

His-HDX-MS has also been utilized by Cebo and colleagues to identify phosphorylated His residues in proteins [60]. They verified that phosphorylation dramatically slows the rate of C-2 hydrogen HDX, thereby allowing the detection of phosphorylated His residues. Vachet’s group showed that the HDX rates of metal-bound His residues are significantly slower than their free forms [61,62] and demonstrated that His-HDX-MS can be used to identify metal-bound His residues. 

### 5.2. Implication of $pK_{a}$ and Kinetic Parameters in the Context of the Function of His Residues in Proteins

#### 5.2.1. $pK_{a}$ Values and Enzyme Catalysis

The $pK_{a}$ values of His residues can shift appreciably due to interactions with the surrounding functional groups in proteins or substrates. It is widely acknowledged that adjacent groups contributing to higher electron density in the imidazole ring, such as RCOO^−^, RO^−^, RS^−^, elevate the $pK_{a}$ of the imidazole ring, while neighboring groups, which diminish the electron density of the imidazole ring, such as RNH_3_^+^, GdnH^+^, ImH^+^, lead to lowering the $pK_{a}$ of the imidazole ring [63]. There are many His residues reported to have exhibited highly deviant $pK_{a}$ values from the well-acknowledged range of 6–7, suggesting the existence of interactions with ionizable functional groups directly or via hydrogen bond networks [2,64]. The relationship between the acid-base properties of His residues and enzyme catalysis was first noted for catalytic residues His12 and His119 in RNase A in 1961 [65]. Since this study, many researchers have attempted to measure the $pK_{a}$ values of these residues using ^1^H NMR [5,8,9,12,14,17,66,67], His-HTX [15,30,68] and His-HDX-MS [10,26]. For example, the $pK_{a}$ value of His12 changes from 5.8 in the free enzyme to 7.1 in the complex with 3′-cytidine monophosphate (3′-CMP) [26]. A similar but less significant alkaline shift in $pK_{a}$ from 6.3 to 7.0 was observed for His119 [26]. These results suggest that His12 and His119 cooperatively act as an acid and a base in their reversed roles before and after the binding of 3′-CMP, which is in consistent with the most classic catalytic mechanism [65]. A recent perspective on the RNase A reaction mechanism has been summerized by Raines and colleagues [69,70]. Thus, measuring the $pK_{a}$ values of catalytic His residues in enzymes may provide some insights for understanding their catalytic mechanisms. 

#### 5.2.2. Double Sigmoidal Titration Curve: $pK_{a}$ Values in His HDX and NMR

Double sigmoidal curves could be observed for His imidazole groups, which are in close proximity to ionizable groups. In the cases of RNase T_1_ and RNase St, several His residues exhibited double sigmoidal titration curves both in HTX and NMR titration experiments [29,31]. A double sigmoidal titration curve in the plot in Figure 7 infers the existence of a charged group near a His residue. Suppose a pair of Lys and His residues are mutually interacting with each other in equilibrium, and their $pK_{a}$ values differ by more than two pH units [30,71]. In an NMR titration experiment, the inflection points at lower and higher pH values are considered to correspond to the p*K*a of the imidazole group of His residues and the amino group of Lys, respectively. However, in the His-HDX experiment, both inflection points reflect the acid-base equilibria of His residues influenced by the protonation/deprotonation of the Lys amino group. This is because the HDX reaction proceeds solely through the equilibrium involving the imidazolium cation ($Im\cdot H^{+}$) of the His residue (see Figure 2). Consequently, the HDX microscopic dissociation constants obtained through the His-HDX experiment—namely p*K*_1_ and p*K*_4_ in Figure 7—reflect the dissociations of the imidazolium group when Lys is protonated (equilibrium between systems I and II) and deprotonated (equilibrium between III and IV), respectively (refer to Figure 3). Although the dissociation constants *K***_2_** and *K***_3_** are not measurable by His-HDX-MS, they can be calculated, if necessary, from the macroscopic dissociation constants obtained from the double sigmoidal curve of the NMR titration experiment, as described by Rabenstein and Sayer [72]. Thus, double sigmoidal titration curves provide two well-defined sets of microscopic $pK_{a}$ and $k_{\varphi }^{max}$ values reflecting the ionizable group(s) around His residues.

### 5.3. His HDX Rate and Solvent Accessibility as Factors Reflecting Protein Structure

The PF values for the His imidazole groups defined in Equation (23) have been demonstrated to correlate well with the solvent-accessible surface area (ASA) calculated from the crystal structures of proteins, as discussed in Section 2.3. Thus, PF values are a useful indicator for predicting the solvent accessibility of His residues in proteins. In the case of an enzyme DHFR in apo and its complex with folate and NADP^+^, the PF values determined for four out of five His residues had a fairly good correlation with ASA, as shown in Figure 8 [28]. Notable exceptions include the anomalous shift in ASA-PF data obtained for His149, which appeared to be shielded from access to the solvent in the apo form more than in the complex, given the approximately two-fold decrease in ASA value in the apo form. However, the PF values of both forms were almost indistinguishable. This inconsistency could possibly be attributed to the difficulty in calculating ASA based on the PDB data of apo-DHFR (5DFR), in which the local structure involving His149 (His149(a) in Figure 8) in the G-H loop fluctuates minutely between “open”, “closed” and “occluded” conformations [73,74]. In contrast, the most appreciable deviation of the PF value from ASA was observed for His114 in DHFR complexed with folate and NADP^+^. According to the X-ray structure of the DHFR–folate–NADP^+^ complex (PDB ID: 1RX2) [74], the involvement of His114, Glu154 and Ser135 in a network of hydrogen bonds could be the main reason for suppression of the HDX reaction of His114 in comparison with the almost unrestrained condition of this residue in the apo form of the enzyme (PDB ID: 5DFR). Thus, the PF–ASA plot may exhibit His residues with unusual structural features beyond solvent accessibility. 

His residues with extremely low $k_{\varphi }^{max}$ values are generally considered to be almost completely shielded from access to water or deeply “buried” within the protein structure, with the exception of His residues coordinating with metal ions. Indeed, the reaction rate of His HDX depends proportionally on the concentration of the deuteroxide ion, $\left[{OD}^{-}\right]$, in bulk water, as shown in Equation (4). However, it is not rare to encounter a fairly high value of ASA being calculated for His residues for which the HDX rates are unusually slow [3]. One notable example includes His25 in FK506 binding protein. The ASA value of His25 in this protein is quite high (52.6% of fully exposed residue). However, the kinetic constants could not be determined under ^1^H NMR titration, likely due to its exceptionally low $pK_{a}$ value [3,75]. The challenge in obtaining a sigmoidal curve is anticipated in Equation (22), explaining that the intrinsic HDX reaction rate, kφmaxi, decreases logarithmically with the reduction in $pK_{a}$ value. 

Table 1 lists several His residues—including His25 in FK506 binding protein—in which the determination of $pK_{a}$ values was challenging due to extremely slow HDX rates and the absence of substantial pH dependency in titration data. His48 in RNase A serves as another example, illustrating a low $pK_{a}$ value accompanied by an exceptionally slow HDX rate. Within the hydrogen bond network, as revealed by the X-ray structure of RNase A (PDB ID: 1RPH), the imidazole group of His48 in β-strand 1 is situated between the hydroxy group of Thr82 in β-strand 1 and the amide carbonyl group of Asp14 in loop 1, as depicted in Figure 4. Within this hydrogen bond network, the imidazole group of His48 is predominantly found in the neutral form, rendering it incapable of participating in HDX reaction [26]. As a result, the HDX rate is significantly slower than the rate estimated by its ASA. Thus, the relationship between PF and ASA values is often not straightforward and requires careful interpretation. However, it could include valuable information, which cannot be obtained using other structural biology techniques.

## 6. Conclusions

In summary, His-HDX-MS offers valuable insights into two key properties of histidine imidazole groups within proteins. First, it provides information on $pK_{a}$ values, reflecting the electrostatic environment surrounding a specific His imidazole group. Second, it can measure the His imidazole group’s HDX rate, which serves as an indicator of solvent accessibility, conformational stability and fluctuation. These parameters are instrumental in understanding enzyme mechanisms, ligand-induced conformational changes in proteins and various other crucial processes. Although there remains a challenge in elaborating on the mechanism of enhancing or attenuating His HDX rate by the environment—influencing the reactivity of the imidazolium group through hydrogen bond or electrostatic interaction—we believe there are ample opportunities for this technique to prove highly beneficial in the field of structural biology research.

His-HDX-MS is well suited to analyzing complex protein mixtures, such as cell lysates. Owing to the slow back exchange of imidazole C-2 D to C-2 H, proteases with strict substrate specificities (e.g., trypsin) can be used, which provides higher peptide discrimination power in peptide identification. Additionally, slow back exchange allows the employment of long chromatographic separation in the LC-MS/MS analysis in order to enhance peptide separation. Additionally, a highly efficient histidine-containing peptide enrichment method is available [34], which significantly simplifies proteome samples and enhances the identification of His-containing peptides. For these reasons, His-HDX-MS should be applied more to proteome-scale samples in the near future, as also noted by Courouble and colleagues in their recent review paper [44]. Lastly, there is currently no computational tool available for automating the data analysis step. The availability of such software will encourage scientists in the field to use this method.

## Data Availability

Not applicable.

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
