# Peer review of "Significance of Histidine Hydrogen–Deuterium Exchange Mass Spectrometry in Protein Structural Biology"

_biology, 2024, doi:10.3390/biology13010037_

Round 1
Reviewer 1 Report
Comments and Suggestions for Authors
Dear Authors,
Your review entitled “Significance of Histidine Hydrogen-Deuterium Exchange Mass Spectrometry in Protein Structural Biology” contains a careful analysis of valuable literature data on the protonation of histidine residues of proteins. This analysis seems to be useful for the researchers investigating the protein structure, function, and dynamics. I recommend to publish this review after consideration of some minor remarks:
1. Keywords (line 34) Instead of “pKa“ must be ”histidine protonation”, proteins.
2. line 63 After….., roughly 4-9, depending ……. environment [ ] (must be references).
3. line 164, equation 2-3, what is Kφ must be indicated.
4. line 473 ……..by Rabenstein and Sayer instead of [49] must be [50].
5. The part 2.3 seems to be overloaded with formulas.
Author Response
Comment 1: Keywords (line 34) Instead of “pKa“ must be ”histidine protonation”, proteins.
We have replaced 'pKa' with 'histidine protonation,' as suggested by the reviewer. We have also added 'acid dissociation constant' to the keywords and replaced “structural biology’ with ‘protein structural biology’.
Comment 2: line 63 After….., roughly 4-9, depending ……. environment [ ] (must be references).
The following three references have been cited as reference numbers 2-4 in the revised manuscript.
- Antosiewicz, J.; Mccammon, J.A.; Gilson, M.K. The Determinants of pKas in Proteins. Biochemistry 1996, 35, 7819–7833.
- Edgcomb, S.P.; Murphy, K.P. Variability in the pKa of Histidine Side-Chains Correlates with Burial within Proteins. PROTEINS: Struct. Funct. Genet. 2002, 49, 1–6.
- Couch, V.; Stuchebrukhov, A.; Histidine in Continuum Electrostatics Protonation State Calculations. Proteins. 2011, 79, 3410-3419.
Comment 3: line 164, equation 2-3, what is kφ must be indicated.
has been defined after Equation 2-3 as follows: ‘where is the pseudo-first order rate constant.’
Comment 4: line 473 ……..by Rabenstein and Sayer instead of [49] must be [50].
We appreciate the reviewer for noticing the error in the citation. We have corrected the error.
Comment 5: The part 2.3 seems to be overloaded with formulas.
We acknowledge the reviewer's concern; however, we believe that these equations play a crucial role in aiding readers' comprehension of the His-HDX mechanism and in understanding how the HDX rate is obtained from mass spectra. Consequently, we have opted not to reduce the number of equations.
Reviewer 2 Report
Comments and Suggestions for Authors
This comprehensive and well-written review highlights a powerful technique that occupies an often overlooked niche in the structural biology field. The authors do an excellent job of describing the theory and methodology of His-HDX-MS and provide a number of insightful examples. I have no issues with the content nor the organization of this review. I am a little curious as to whether unusual exchange values are observed for His residues that are supposed to engage in low-barrier hydrogen bonds, as is theorized for serine proteases. If there is anything interesting there, the authors could perhaps comment on it. Otherwise, I have only a few very minor edits to request.
1. Figure 2 legend: the word "residue" at the end appears to be a typo.
2. Page 4 line 122: "radio activities" should be one word.
3. Page 10 line 319: "His-HDH-MS" should read "His-HDX-MS".
4. Page 10 line 333: "Alternatively way" should read "An alternate way"
5. Page 11 line 380: The text references "a study by Hayashi and colleagues" but there is no reference number.
6. Page 11 line 394: "prevents" should be "prevent".
7. Page 12 line 445: There appears to be a typo here in "pKages". I assume the authors intended to write "measure the pKa changes from 5.8..." or something similar.
8. Page 13, line 457: delete "a" to read "exhibited double sigmoidal titration curves".
9. Page 13 line 474: "provides" should be "provide".
10. Page 14, line 498: "Thus" should begin a new sentence to read "Section 2.3. Thus, PF values...".
11. Page 14 line 505 is a sentence fragment. Consider revising to "However, the PF values of both forms were almost indistinguishable."
12. Page 15 line 528: Change to "A notable example includes His25..."
13. Page 16 line 564: Change "serves as indicators" to "serves as an indicator".
Author Response
Major comment: I am a little curious as to whether unusual exchange values are observed for His residues that are supposed to engage in low-barrier hydrogen bonds, as is theorized for serine proteases. If there is anything interesting there, the authors could perhaps comment on it.
We appreciate the insightful comment from the reviewer. It is intriguing to observe the impact of HDX on His residues involved in low-barrier hydrogen bonds. However, the application of the His-HDX-MS technique to His residues participating in such bonds has not been demonstrated. Therefore, we currently refrain from making any comments on this aspect. We have included the following statement in the conclusion section.
‘Although there remains a challenge to elaborate on the mechanism of enhancing or attenuating the His HDX rate by the environment, influencing the reactivity of the imidazolium group through hydrogen bond or electrostatic interaction, we believe there are ample opportunities for this technique to prove highly beneficial in the field of structural biology research.’
Comment 1: Figure 2 legend: the word "residue" at the end appears to be a typo.
We have corrected the error.
Comment 2: Page 4 line 122: "radio activities" should be one word.
We have corrected the typo.
Comment 3: Page 10 line 319: "His-HDH-MS" should read "His-HDX-MS"
We have corrected the typo.
Comment 4: Page 10 line 333: "Alternatively way" should read "An alternate way"
We have corrected it.
Comment 5: Page 11 line 380: The text references "a study by Hayashi and colleagues" but there is no reference number.
The following paper has been cited as reference# 26.
- Hayashi, N.; Kuyama, H.; Nakajima, C.; Kawahara, K.; Miyagi, M.; Nishimura, O.; Matsuo, H.; Nakazawa, T. Imidazole C-2 Hydrogen/Deuterium Exchange Reaction at Histidine for Probing Protein Structure and Function with Matrix-Assisted Laser Desorption Ionization Mass Spectrometry. Biochemistry 2014, 53, 1818–1826
Comment 6: Page 11 line 394: "prevents" should be "prevent".
We have corrected the typo.
Comment 7: Page 12 line 445: There appears to be a typo here in "pKages". I assume the authors intended to write "measure the pKa changes from 5.8..." or something similar.
Some of the sentence was accidentally deleted, but we've reinstated the missing part. The complete sentence is provided below.
‘Since this study, many researchers attempted to measure the values of these residues by 1H NMR [5,8,9,12,14,17,66,67], His-HTX [15,30,68], and His-HDX-MS [10,26]. For example, the value of His12 changes from 5.8 in the free enzyme to 7.1 in the complex with 3′-cytidine monophosphate (3′-CMP) [26].’
Comment 8: Page 13, line 457: delete "a" to read "exhibited double sigmoidal titration curves".
We have corrected the error.
Comment 9: Page 13 line 474: "provides" should be "provide".
We have corrected the typo.
Comment 10: Page 14, line 498: "Thus" should begin a new sentence to read "Section 2.3. Thus, PF values...".
We have divided the sentence into two parts, as the reviewer suggested, as follows.
‘The PF values for His imidazole groups defined by Equation 2-21 have been demonstrated to correlate well with the solvent-accessible surface area (ASA) calculated from the crystal structures of proteins, as discussed in Section 2.3. Thus, PF values are a useful indicator for predicting the solvent accessibility of His residues in proteins.’
Comment 11: Page 14 line 505 is a sentence fragment. Consider revising to "However, the PF values of both forms were almost indistinguishable."
We have changed the sentence as the reviewer suggested.
Comment 12: Page 15 line 528: Change to "A notable example includes His25..."
We have changed the sentence as the reviewer suggested.
Comment 13: Page 16 line 564: Change "serves as indicators" to "serves as an indicator".
We have changed the sentence as the reviewer suggested.
Reviewer 3 Report
Comments and Suggestions for Authors
In the current manuscript, the authors have described the importance of His-HDX measurements in estimating the pKa of the histidine imidazole groups and conformational dynamics in a protein. The review is quite thorough and is acceptable for publication. However, the authors should address the following concern.
1. Lines 463-467: In this section, the authors have alluded to the fact that the two inflection points in the titration curve in His-HDX experiment, correspond to the pKa values of the His residues, as the equilibrium is dominated by the formation of imidazolium cation. However, during sample preparation for HDX, the side chain of the Lys residues will remain protonated, and these are expected to perturb the ionization of the neighboring histidine residues. In that case, one of the inflection points in the titration curve should correspond to pKa value of the Lys residue. Can the authors elaborate on this?
Comments on the Quality of English LanguageListed below are a few typographical errors the authors should correct.
a. Line 302: Correct ‘remained’ to ‘remaining’.
b. Line 339: Correct ‘prteases’ to ‘proteases’
c. Line 365: Correct ‘chramoatographic’ to ‘chromatographic’.
d. Lines 496 – 499: Split into 2 sentences after Section 2.3.
Author Response
Comment 1: Lines 463-467: In this section, the authors have alluded to the fact that the two inflection points in the titration curve in His-HDX experiment, correspond to the pKa values of the His residues, as the equilibrium is dominated by the formation of imidazolium cation. However, during sample preparation for HDX, the side chain of the Lys residues will remain protonated, and these are expected to perturb the ionization of the neighboring histidine residues. In that case, one of the inflection points in the titration curve should correspond to pKa value of the Lys residue. Can the authors elaborate on this?
In NMR titration experiment, the inflection points at the lower and higher pH correspond to the of the imidazole group of His and the amino group of Lys, respectively. However, in the His-HDX experiment, both two inflection points reflect the acid-base equilibria of His residues influenced by the protonation/deprotonation of Lys amino group because the HDX reaction proceeds solely through the equilibrium involving the imidazolium cation () of the His residue. We have clearly stated this point as follows in the text under 5.2.2.
‘In an NMR titration experiment, the inflection points at lower and higher pH values are considered to correspond to the pKa of the imidazole group of His residues and the amino group of Lys, respectively. However, in the His-HDX experiment, both inflection points reflect the acid-base equilibria of His residues influenced by the protonation/deprotonation of the Lys amino group. This is because the HDX reaction proceeds solely through the equilibrium involving the imidazolium cation () of the His residue (see Scheme 2). Consequently, the HDX microscopic dissociation constants obtained through the His-HDX experiment, namely pK1 and pK4 in Figure 7, reflect the dissociations of the imidazolium group when Lys is protonated (equilibrium between systems I and II) and deprotonated (equilibrium between III and IV), respectively (refer to Scheme 3).’
Comment 2: typos
- Line 302: Correct ‘remained’ to ‘remaining’.
- Line 339: Correct ‘prteases’ to ‘proteases’
- Line 365: Correct ‘chramoatographic’ to ‘chromatographic’.
- Lines 496 – 499: Split into 2 sentences after Section 2.3.
All these typos have been corrected.
Reviewer 4 Report
Comments and Suggestions for Authors
The review of Miyagi and Nakazawa deals with the application of hydrogen-deuterium exchange mass spectrometry (HDX-MS) to the measurement of pKa values and solvent accessibilities of His residues (His-HDX-MS) in a protein. Indeed, as commented in the review the His residue is a key amino acid in structural biology, in particular for its involvement in hydrogen bonding, in catalysis and metal binding. The electrostatic environment of a His residue changes the pKa of its imidazole group so that its measurement can give important structural information on the protein as well as it allows to analyse structural changes induced by the interaction with other proteins or ligands. The review is well written and it follows a logical order of the topics: an introduction on the role of His in structural biology, an overview of techniques that can be used to measure the pKa values of His residues and a detailed description of the His-HDX-MS technique and of its applications. Given that His-HDX-MS is an interesting structural biology technique and that it was developed also with the contribution of Dr. Dealwis, this Reviewers considers that it is certainly suitable for publication in the Special Issue "Hybrid Methods for Structural Biology and Drug Design: A Memorial Issue for Dr. Chris G. Dealwis" of Biology journal. However, before publication this Reviewer suggests minor revisions to be addressed and that are outlined in the following:
- Line 78: “..developed in 2008”. Add the reference of the paper the described the development of His-HDX-MS
- Line 132: “..mass spectrometry.” Add the reference.
- Line 271: “..the HTX method..”. Write also the extended name of HTX to explain the acronym.
- Line 303: “..immobilised proteases.” This reviewer suggests to add the pH of the digestion also at this level of the text to underline differences with the amide-HDX-MS workflow. Moreover, it would be useful to mention if the His-HDX-MS protocol includes a quenching step or pH adjustment step before protease digestion since the exchange is performed at various pD values.
- Line 334: “..(CID)”. In amide-HDX-MS, the CID fragmentation causes scrambling of exchanged D. Add a comment on this aspect for the His-HDX-MS technique.
- Line 365: “..chromatographic separation..”. In amide-HDX-MS, the LC-MS separation of the peptides is performed close to 0°C to avoid back exchange but this condition affects the separation performance. Add a comment on the temperature of the LC-MS analysis in His-HDX-MS.
- Line 403: ...unlike amide-HDX-MS..”. Since amide-HDX-MS and His-HDX-MS allow to obtain different structural information, maybe the comment can be more focused on the complementarity of the two techniques.
Minor corrections of English language mainly regard word spelling.
Author Response
Comment 1: Line 78: “..developed in 2008”. Add the reference of the paper the described the development of His-HDX-MS
The following reference has been added.
Miyagi, M.; Nakazawa, T. Determination of pKa Values of Individual Histidine Residues in Proteins Using Mass Spectrometry. Anal Chem 2008, 80, 6481–6487
Comment 2: Line 132: “..mass spectrometry.” Add the reference.
The following reference has been added.
Miyagi, M.; Nakazawa, T. Determination of pKa Values of Individual Histidine Residues in Proteins Using Mass Spectrometry. Anal Chem 2008, 80, 6481–6487
Comment 3: Line 271: “..the HTX method..”. Write also the extended name of HTX to explain the acronym.
‘hydrogen-tritium exchange’ has been added.
Comment 4: Line 303: “..immobilised proteases.” This reviewer suggests to add the pH of the digestion also at this level of the text to underline differences with the amide-HDX-MS workflow. Moreover, it would be useful to mention if the His-HDX-MS protocol includes a quenching step or pH adjustment step before protease digestion since the exchange is performed at various pD values.
As suggested, the pH for the digestion step has been added. Also, the following sentence has been added.
‘After the incubation, the reaction is quenched by adding formic acid to adjust the solution’s pH to below 4.’
Comment 5: Line 334: “..(CID)”. In amide-HDX-MS, the CID fragmentation causes scrambling of exchanged D. Add a comment on this aspect for the His-HDX-MS technique.
The following sentences have been added in Section 3.
‘These examples illustrate the absence of C2-deuterium migration to other peptide regions during CID or ETD. This stands in contrast to amide-HDX-MS, where notable scrambling of amide-deuterons occurs during CID [36,37], albeit to a much smaller extent during ETD [38–40] or UVPD (ultraviolet photodissociation) [41,42].
Comment 6: Line 365: “..chromatographic separation..”. In amide-HDX-MS, the LC-MS separation of the peptides is performed close to 0°C to avoid back exchange but this condition affects the separation performance. Add a comment on the temperature of the LC-MS analysis in His-HDX-MS.
We appreciate the reviewer's suggestion and have incorporated the following sentences, citing Guillarme’s and Lundanes’s papers.
‘Additionally, in amide-HDX-MS, the LC-MS separation of peptides is conducted at low temperature (close to 0 °C) to avoid back exchange. However, since essentially no back-exchange reaction occurs below pH 4 [10], in His-HDX-MS, a long chromatographic separation of peptides at a higher temperature can be employed to enhance the efficiency of peptide separation [48,49].
Guillarme, D.; Heinisch, S.; Rocca, J.L. Effect of Temperature in Reversed Phase Liquid Chromatography. J. Chromatogr. A 2004, 1052, 39–51
Lundanes, E.; Greibrokk, T. Temperature Effects in Liquid Chromatography. Adv. Chromatogr. 2006, 44, 45–77
Comment 7: Line 403: ...unlike amide-HDX-MS..”. Since amide-HDX-MS and His-HDX-MS allow to obtain different structural information, maybe the comment can be more focused on the complementarity of the two techniques.
We appreciate the reviewer’s insightful comments. The fourth paragraph in Section 5.1 highlights the complementary nature of both methods. The paragraph is provided below.
‘Surewicz’s group has employed His-HDX-MS in conjunction with amide-HDX-MS to study the structures of different forms of prion proteins [54–57]. These studies have showcased the complementary nature of His-HDX-MS and amide-HDX-MS. The former delves into the microenvironment of His residues, while the latter unveils the conformational flexibility of backbone amides throughout the structure. Therefore, it is recommended to leverage both techniques when technical expertise is available.’
Reviewer 5 Report
Comments and Suggestions for Authors
The articles delivered a nice piece of work, extensively describe the HDX-MS advantage over other mass spec techniques in assigning the His residue, which would be helpful to understand the protein conformation and structural properties. Also helpful for the readers to identify the bound structural water molecules.
We could see some minor mistakes in the manuscript.
The scheme equation 1 is not a complete figure.
Which enzyme will be taken to digest only the water molecules (Fig.6).
In Table 1, It would be great if the author provides RNAs and other proteins uniport id and Histidine residue position instead of writing His residue.
Author Response
Comment 1: The scheme 1 is not a complete figure.
The figure is now complete.
Comment 2: Which enzyme will be taken to digest only the water molecules (Fig.6).
We have modified the figure. Trypsin has been indicated in the figure as a possible protease to be used.
Comment 3: In Table 1, It would be great if the author provides uniport id and Histidine residue position instead of writing His residue.
We have added Protein Data Bank IDs, as we believe they are more useful for the readers than UniProt IDs. Additionally, we have replaced one of the headings of the table ‘Histidine residues’ with ‘Positions of His Residues’.